# The Relationship between Nutritional Risk and the Most Common Chronic Diseases in Hospitalized Geriatric Population from Central Poland

**DOI:** 10.3390/nu15071612

**Published:** 2023-03-27

**Authors:** Serena S. Stephenson, Agnieszka Guligowska, Anna Cieślak-Skubel, Agnieszka Wójcik, Ganna Kravchenko, Tomasz Kostka, Bartlomiej K. Sołtysik

**Affiliations:** Department of Geriatrics, Healthy Ageing Research Centre (HARC), Medical University of Lodz, Haller Sqr. No. 1, 90-647 Lodz, Poland

**Keywords:** older people, malnutrition, undernutrition, NRS-2002, concomitant diseases

## Abstract

The aim of this study was to assess the relationship between Nutrition Risk Screening 2002 (NRS-2002) and the prevalence of concomitant chronic diseases among hospitalized older adults. This study included 2122 consecutively hospitalized older participants with an average age of 82 years. The criteria to participate were the ability to communicate and give consent. In multivariate design, the prevalence of nutritional risk with at least 3 points in the NRS-2002 score was associated with the presence of stroke, atrial fibrillation, dementia and pressure ulcers. Patients with arterial hypertension, lipid disorders, osteoarthritis and urine incontinence had a significantly lower (better) NRS-2002 score. The explanation of the inverse relationship between some disorders and nutritional risk may be their occurrence in relatively earlier age and the relationship with body mass index. In conclusion, the study revealed which medical conditions coexist with the increased nutritional risk in a “real-world” hospitalized geriatric population. The hospital admission of an older subject with stroke, atrial fibrillation, dementia or pressure ulcers should primarily draw attention to the nutritional risk of the patient.

## 1. Introduction

The term malnutrition is often used to describe a deficiency in nutrition that causes adverse effects on the body and its normal functions [1]. Malnutrition in hospitalized patients represents a heavy healthcare burden worldwide [2,3]. An increase in malnutrition-related diseases in people with multiple comorbidities is a growing health concern, and it is strictly related to both the aging of the general population and the improvement in healthcare [4]. Malnutrition adversely affects physical well-being, interferes with treatment, and increases the duration of hospital stay [5]. Improvements in nutrition are known to bring tangible benefits to older people and many age-related diseases and conditions can be prevented, modulated or ameliorated by good nutrition [6]. In order to develop an appropriate nutritional plan and intervene immediately, the screening of patients’ nutritional status is normally performed upon hospital admission [7].

The determinants of malnutrition are especially important in the most vulnerable older population, with the risk of naturally developing general poor health or chronic diseases [8]. Some of these factors could be potentially influenced by the environment of the geriatric population, which has been shown to have a significant impact on nutrition [9]. If an older person is residing in a hospital or long-term care facility, it has been shown that they are more likely to have a poor nutrition status compared to a community-dwelling older person [10]. Indeed, some investigators have shown that up to 60–80% of European geriatric hospital patients are malnourished [10,11]. Several factors contribute to the worsening of nutritional status during hospitalization: illness-related loss of appetite, fasting for diagnostic procedures, drug-related side effects, diseases that compromise the regular functioning of the digestive system and the poor patient management [12].

For this purpose, numerous screening tools have been developed and validated [8,13]. An effective nutritional screening tool must be practical, i.e., those who are going to use the tool must find it rapid and simple, and such a tool must also have high validity and reliability [14]. The Nutritional Risk Screening (NRS-2002) is a validated tool for medical services to identify malnourished patients who would benefit from nutritional intervention [14,15]. It may predict the probability of a better or worse outcome due to nutritional factors, but also whether nutritional treatment is likely to influence the clinical prognosis [16]. Unlike other nutritional assessment tools that mainly focus on laboratory indicators such as the albumin or lymphocyte count, NRS-2002 takes the effect of the changes in food intake and disease severity into consideration [17,18]. Nowadays, NRS-2002 stands out as an effective, flexible, and comprehensive nutritional assessment tool and has been extensively used in the clinical nutrition practice to provide nutritional information [19,20].

The severity of concomitant diseases is the basic determinant of overall NRS-2002 score [18]. While the severity of acute disease or surgery may be quite reliably assessed, the contribution of concomitant chronic diseases to overall NRS-2002 is less clear. In clinical practice, the influence of the most common accompanying diseases should be appropriately weighed. In the currently available literature, there is a relatively wide range of papers comparing NRS-2002 with specific individual disorders [21,22,23,24,25] or with different nutritional tools [26,27]. However, there is an insufficiency of research relating to nutritional risk and the variety of age-related diseases compared concomitantly. Hence, the aim of this study was to assess the relationship between NRS and the prevalence of concomitant chronic diseases among hospitalized older adults.

## 2. Materials and Methods

### 2.1. Design of the Study and Participants

The study population that was selected consisted of older adults, aged 60 years old and above, who were hospitalized in the Geriatric Department, Central Veterans Hospital located in Lodz, Poland. Patients were recruited from January 2012 to December 2019. During the period 2020–2022, the department served partially as a COVID-19 ward. The total number of individuals was selected with the following inclusion criteria: admission to the department, aged 60 years and above, ability to communicate efficiently, complete data, and giving informed consent. After screening, 2122 patients (631 men and 1491 women) who met the criteria were enrolled into the analysis.

### 2.2. Nutritional Questionnaire

NRS-2002 was designed as a tool to identify patients at nutritional risk. It assesses risk through two criteria: impaired nutritional status and disease severity [15]. Nutritional status was determined by variables which comprised age, body mass index (BMI), reduced dietary intake and recent body mass loss. BMI was calculated by dividing the weight (in kilograms) by height squared (in meters). The NRS-2002 score consisted of the total nutritional score (impaired nutritional status), the severity of disease score (increase in nutritional requirement) and the age adjustment score (<70 or ≥70 years). The total number of points scales from 0 to 7. Patients with a score of 3 or more are suggested to be nutritionally at risk.

### 2.3. Concomitant Diseases

The prevalence of diseases such as arterial hypertension, diabetes, lipid disorders, current or previous stroke, coronary artery disease, current or previous myocardial infarction, atrial fibrillation, heart failure, obstructive lung diseases (chronic obstructive pulmonary disease and asthma), osteoarthritis, osteoporosis, current or previous fracture, gastrointestinal diseases, neoplastic diseases, depression, dementia, pressure ulcers and urine incontinence was scrutinized from the patients’ database. The vast majority of subjects presented with multiple comorbidities; however, the presented diseases were analyzed separately.

### 2.4. Statistical Analysis

The normality of distribution was analyzed using a Shapiro–Wilk test. As several variables were not normally distributed, data were expressed both as the mean ± standard deviation (SD) and median (25–75% quartiles). The quantitative variables were compared using a Mann–Whitney U-test and qualitative variables using a chi-square test. Spearman correlation coefficients were used to calculate the relationship between the two quantitative variables. The results of NRS-2002 were presented in two different ways: for patients’ characteristics as a raw variable and for logistic regression NRS-2002 was dichotomized as <3 and ≥3 points. Multivariate ordinal logistic regression analysis was further used to investigate the relationship between concomitant diseases and NRS-2002 (≥3 points against <3 points). Statistical significance was set at *p* ≤ 0.05. Statistical analysis was performed using Statistica 13.1.

### 2.5. Ethical Certification

This study was conducted according to the guidelines of the Declaration of Helsinki, and approved by the Ethics Committee of the Medical University of Lodz with the approval number RNN/300/17/KE. Patients signed informed consent for all the diagnostic and therapeutic procedures during hospitalization. All of the gathered data were confidential.

## 3. Results

The mean age for the whole study populations was 82.18 ± 7.90 years. Table 1 shows the characteristics of 2122 patients according to gender. Men had a higher body mass compared to the women. The prevalence of diabetes, myocardial infarction, atrial fibrillation and neoplastic disease was higher in men than in women. The prevalence of arterial hypertension, lipid disorders, osteoarthritis, osteoporosis, fractures, depression, dementia and urine incontinence was higher in women than in men.

Table 2 shows the comparison of NRS-2002 scores according to the prevalence of concomitant diseases. The presence of arterial hypertension (in men), lipid disorders, osteoarthritis, osteoporosis (in women) and urine incontinence was associated with significantly lower (better) NRS-2002 scores in bivariate analyses. The presence of stroke, atrial fibrillation, heart failure, neoplastic diseases (in men), dementia and pressure ulcers was associated with significantly higher (worse) NRS-2002.

Additional bivariate analyses were performed between the presence of diseases and age (Table 3). The patients presenting with diseases such as arterial hypertension (women), coronary artery disease, myocardial infarction, atrial fibrillation, heart failure, osteoporosis (men), fractures (women), neoplastic diseases (men), dementia and pressure ulcers were significantly older, whereas the patients with lipid disorders and gastrointestinal diseases (women) were significantly younger in comparison to those without disease.

In order to simultaneously assess an association of accompanying diseases to NRS-2002, logistic regression was performed. For the whole study group, the analysis is shown in Figure 1. Diseases which expressed significance in bivariable analysis were employed in multivariable analysis. Diseases such as stroke, atrial fibrillation, dementia, and pressure ulcers were independently associated with increased values of NRS-2002 (increased odds ratio for nutritional risk). Arterial hypertension, lipid disorders, osteoarthritis and urine incontinence were associated with lower values of NRS-2002.

Figure 2 shows the simultaneous statistical impact of concomitant diseases on NRS-2002 scores in women. Stroke, atrial fibrillation, dementia and pressure ulcers were related to higher values of NRS-2002 (increased odds ratio for nutritional risk). Arterial hypertension, lipid disorders, and urine incontinence were related to lower values of NRS-2002.

Figure 3 shows the simultaneous statistical impact of concomitant diseases on NRS-2002 in men. Dementia and pressure ulcers were related to higher values of NRS-2002 (increased odds ratio for nutritional risk). Lipid disorders, osteoarthritis and urine incontinence were related to lower values of NRS-2002.

Although NRS-2002 is connected with BMI (as contains BMI in its definition), some further adjustments for the four different levels of BMI were performed. For BMI cut-off points of 23, 25, 27 and 30 kg/m^2^, the positive relationship between these two variables was extremely strong (chi-square *p* < 0.001 for all BMI cut-offs against dichotomized NRS-2002).

When BMI was entered into the regression models, it was such a strong predictor of NRS-2002 that only some diseases were still present in the model. Pressure ulcers, atrial fibrillation and dementia (borderline significance) were related to a higher nutritional risk, while lipid disorders, urine incontinence and osteoarthritis (only in one model) were related to lower nutritional risk.

## 4. Discussion

Presented research focuses on the association between NRS-2002 and a variety of concomitant diseases among older people. To the best of our knowledge this is the first study concurrently assessing the relationship between nutritional risk and the most common chronic diseases in a large hospitalized geriatric population. Our data indicate that this relationship is complex and inhomogeneous—not all concomitant disorders are associated with worse nutritional status. The results indicate that there is a bidirectional association between nutritional risk and disorders typical for hospitalized geriatric patients. The presence of stroke, atrial fibrillation, dementia and pressure ulcers are associated with increased risk of malnutrition. On the other hand, lipid disorders, arterial hypertension, osteoarthritis or urine incontinence are statistically bounded with less nutritional risk.

The screening and assessment of malnutrition, also with NRS-2002, have proven their predictive values in different populations of patients [14,15,20,21,25,28,29,30]. In several studies, either targeting specific patients or performing in more general setting, malnutrition has been linked to diverse clinical conditions: age, gender, low BMI, infections, cancer, diabetes mellitus, grade of renal function, acute kidney injury, pulmonary diseases, gastrointestinal disorders, depression, cognitive/functional geriatric tests, or generally high comorbidity and polypharmacy [31,32,33,34,35,36,37].

In the present study, we investigated the relationship between NRS-2002 at admission and the presence of chronic diseases among hospitalized older adults. The prevalence of nutritional risk with at least three points in the NRS-2002 score was associated with the presence of stroke, atrial fibrillation, heart failure (in bivariate analyses), dementia, and pressure ulcers. All these conditions are closely related to advancing age. Malnutrition is frequently observed in patients with stroke [38]. Patients with ischemic stroke at risk of malnutrition are more likely to develop infection complications than those with normal nutrition [39]. The nutritional status can affect the occurrence of new-onset atrial fibrillation in acute myocardial infarction patients [40]. Atrial fibrillation is related to obesity with long-term increased incidence independently of other risk factors [41,42]. Our findings suggest that malnutrition may be also related to the higher prevalence of atrial fibrillation and this fraction of patients seems to be significantly older. The incidence of complications and the median length of the hospital stay are significantly higher in heart failure patients at nutritional risk [43]. In a retrospective study including 2830 heart failure patients, a high NRS-2002 score was strongly and independently associated with the incidence of 1-year re-hospitalization and the length of initial hospital stay [44]. Our results reveal a significantly higher NRS-2002 among patients with heart failure. As for atrial fibrillation, these patients are older as compared to those without the disease.

In dementia, many contributing factors must be considered, including nutrition [45]. Having a cognitive impairment determines malnutrition in successfully ageing populations whilst dementia is reported to be associated with malnutrition within usual and accelerated ageing populations [46]. The prevalence of malnutrition was strongly correlated with the severity of dementia in hospitalized patients [47]. In 4095 geriatric hospital patients, subjects with cognitive dysfunction had a higher NRS-2002 score compared to cognitively intact subjects [48]. Our results comply with previous reports with dementia contributing to a worse NRS-2002 in the whole study population and separately in women and men.

Pressure ulcers are one of the most common occurrences in bedridden subjects [49]. There is significantly higher risk for pressure injury in patients who are at risk for undernutrition compared with those who are not at risk [50]. Prolonged immobilization, sensory deficit, circulatory disturbances and poor nutrition have been identified as important risk factors in the development of pressure ulcers formation [51]. Litchford’s et al. study has reported associations between the declining nutrition status and risk for pressure ulcers [52]. The significant association between the NRS-2002 and pressure ulcers in a mixed hospital population was described [53]. Likewise in our study, pressure ulcers, more evident in the oldest hospitalized subjects, were the most powerful condition related to a worse nutritional status.

Interestingly, we found that patients with arterial hypertension, lipid disorders, osteoarthritis, and urine incontinence have a significantly lower NRS-2002. These data contrast with a few previous studies. An elevated Geriatric Nutritional Risk Index was associated with an earlier age of hypertension onset in the older Chinese population [54]. In a cross-sectional study conducted in Sri Lanka with 999 participants aged on average 70.8 years, hypertension, alcohol consumption and increased age were positively associated with malnutrition [55]. In a cross-sectional study involving 330 Thai community-dwelling older adults, factors significantly associated with an increased nutritional risk were: age ≥80 years, low income, living alone, moderate-to-severe pain, dyslipidemia, osteoarthritis, poor physical performance and ≥1 fall in the previous year [56].

The explanation of an inverse relationship of some disorders to nutritional risk found in our study may be their occurrence in relatively earlier age and in relationships with overweight/obesity. In the present study, BMI decreased (rho = −0.23) while the NRS-2002 score increased (rho = 0.20) with age of the patients. NRS-2002 was inversely related to BMI (rho = −0.27) and BMI was a strong and independent determinant of a lower (better) NRS-2002 score in multivariate analyses. Obesity is not a key problem in older hospitalized patients. The prevalence of obesity among adults aged 65–74 years is higher than in those aged 75 and over [57]. Age is an independent risk factor associated with an elevation in lipid levels in middle-age, and is associated with declining lipid levels in older (>55–60) years [58]. The incidence rates of symptomatic osteoarthritis of either hand, or knee or hip rapidly increase around the age of 50 and then level off after age 70 [59,60]. The prevalence of symptomatic knee osteoarthritis was reported to non-linearly increase with age to show the highest annual incidence between the 55th and 64th year of life [61].

In the literature, there are papers indicating a positive correlation between the occurrence of osteoarthritis and obesity [62]. According to Australian research, the prevalence of osteoarthritis is seven-fold higher among obese subjects [63]. Likewise, overweight and obesity are important risk factors for urinary incontinence [64]. The association of hypertension and lipid disorders with obesity is well known [65,66]. Therefore, this relationship to age and BMI may explain our results of better (lower) NRS-2002 scores in patients with those four disorders, while taking into account other comorbidities. All four selected medical conditions are also characterized by their relatively non-debilitating course—when appropriately treated. Lower NRS-2002 scores in all four of these medical conditions do not necessarily mean that these groups of patients are at less nutritional risk. It only shows that, in a “real-world” mixed geriatric hospitalized population, the nutritional impact of these conditions is age- and BMI-dependent.

Several medical conditions highlighted in previous reports were not selected as independent statistical determinants of malnutrition in our analysis. COPD is one of the risk factors for malnutrition among older subjects [67] and, based on some research, the prevalence of malnutrition according to NRS-2002 in COPD patients is as much as 57% [68]. NRS-2002 has been indicated as a tool for predicting 1 year mortality prognosis in patients with COPD [69]. In contrast to arthritis, osteoporosis is strictly linked with undernutrition. It is particularly connected with protein [70] and calcium/vitamin D3 deficiency [71]. A low BMI is an independent risk factor of that condition in the Frax algorithm [72]. Fractures are predominantly linked with older age and malnutrition [73,74]. Additionally, subjects who are malnourished often have negative outcomes after hip fracture [75]. Malnutrition is connected with worse functional recovery, and an increase in mortality after hip fracture [76,77]. Hip fracture patients who are both overweight or obese, and malnourished, have significantly and substantially worse clinical outcomes than their well-nourished—albeit overweight or obese—counterparts [78].

Gastrointestinal diseases are associated with malnutrition as the variety of symptoms and disorders hinders proper nutrition [79,80]. The validity of NRS-2002 in the nutritional assessment was confirmed in disorders such as acute diverticulitis [81] and inflammatory bowel disease [82]. Malnutrition predicts poorer clinical outcomes for people with cancer [83,84]. Malnutrition negatively impacts the quality of life and treatment, and it has been estimated that up to 10–20% of cancer patients die due to consequences of malnutrition rather than for the tumor itself [85]. The prevalence of malnutrition was related to the presence and severity of depression in different groups of patients [34,47]. In the present study, of these medical conditions, heart failure (in both women and men) and neoplastic diseases (in men) were related to a higher NRS-2002, while osteoporosis was linked with a significantly lower NRS-2002 in women, but only in bivariable analyses. These results indicate that in a multimorbid geriatric inpatient population, the relative burden of these conditions may be less pronounced compared to other diseases.

The main strengths of our study are the number of subjects and the assessment of a wide variety of geriatric disorders. Nevertheless, there are several limitations in this study that could be addressed in future research. This study focused on older inpatients in central Poland. The patients had multiple medical problems but not all the concomitant disorders were scrutinized. Larger multicenter studies in different populations would enable one to draw more conclusions. The relationship between nutritional status and concomitant diseases may also be different during long-term hospitalization or in an institutional environment [86]. We only used one short nutritional screening test—NRS-2002. Other nutritional assessment tools might have performed differently. Finally, it is a cross-sectional analysis, monitoring the nutritional status and its predictive value during and after the hospitalization would probably bring new information.

## 5. Conclusions

This study revealed which medical conditions coexist with the increased nutritional risk in a “real-world” hospitalized geriatric population. The hospital admission of older subjects with stroke, atrial fibrillation, heart failure, dementia or pressure ulcers should primarily draw attention to the nutritional risk of the patient.

## Figures and Tables

**Figure 1 nutrients-15-01612-f001:**
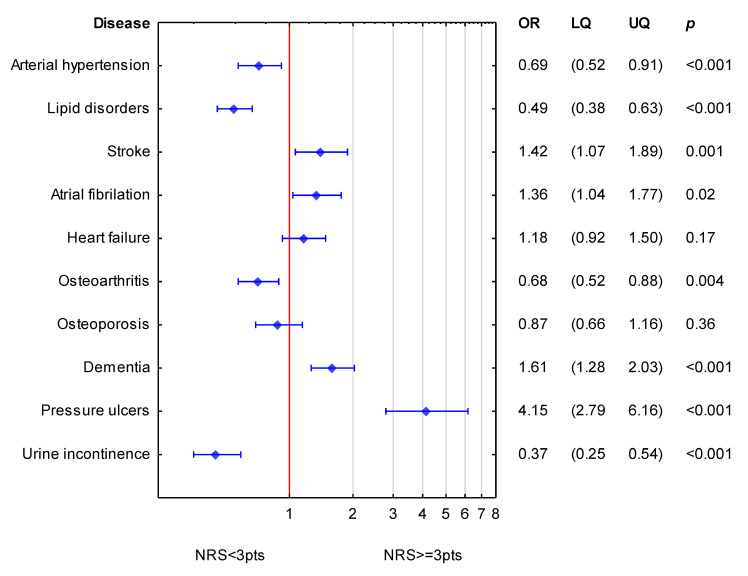
Simultaneous association of diseases accompanying NRS-2002 for the whole group of patients. (OR—odds ratio; LQ—lower quartile; UQ—upper quartile; *p*—*p* value). Reference value (OR = 1; red line) corresponds to the absence of particular disease. Blue symbols correspond to odds ratios and 25–75% confidence intervals for the presence of particular disease.

**Figure 2 nutrients-15-01612-f002:**
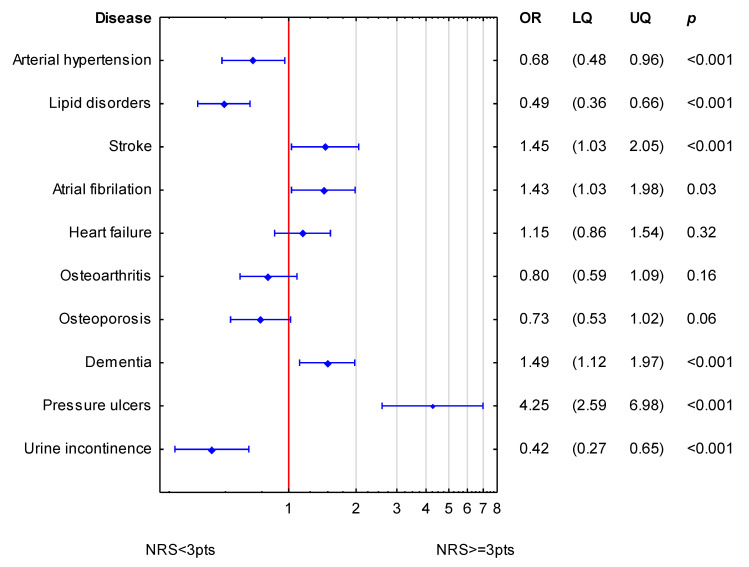
Simultaneous association of accompanying diseases to NRS-2002 in women. (OR—odds ratio; LQ—lower quartile; UQ—upper quartile; *p*—*p* value). Reference value (OR = 1; red line) corresponds to the absence of particular disease. Blue symbols correspond to odds ratios and 25–75% confidence intervals for the presence of particular disease.

**Figure 3 nutrients-15-01612-f003:**
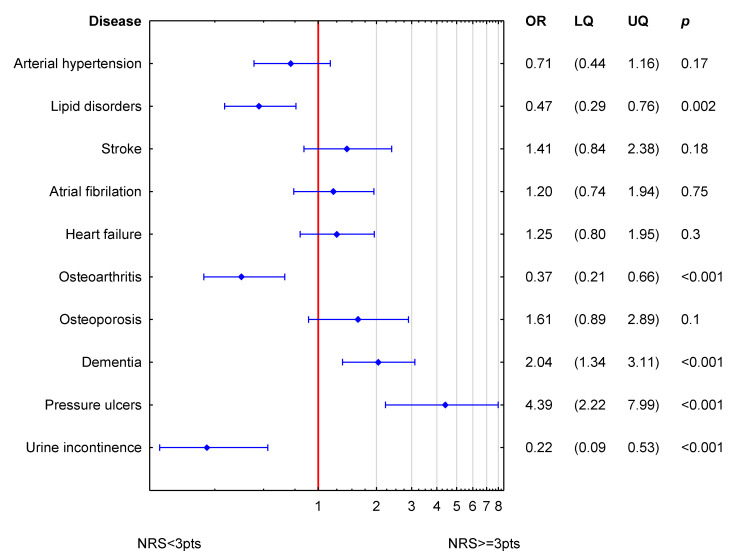
Simultaneous association of accompanying diseases to NRS-2002 in men. (OR odds ratio; LQ—lower quartile; UQ—upper quartile; *p*—*p* value). Reference value (OR = 1; red line) corresponds to the absence of particular disease. Blue symbols correspond to odds ratios and 25–75% confidence intervals for the presence of particular disease.

**Table 1 nutrients-15-01612-t001:** Characteristics of the patients according to gender.

Disease	Women *n* = 1491	Men *n* = 631	*p*-Value
Age; mean ± SD (median and quartiles)	82.33 ± 7.68 83 (78–88)	81.62 ± 8.38 83 (76–88)	*p* = 0.91^(U)^
BMI, m/kg^2^; mean ± SD (median and quartiles)	26.23 ± 5.42 25.5 (22.2–29.3)	26.09 ± 4.86 25.4 (22.9–28.5)	*p* = 0.40^(U)^
Body mass, kg; mean ± SD (median and quartiles)	63.93 ± 14.57 62 (53–72)	74.72 ± 15.39 73 (65–83)	*p* < 0.001^(U)^
NRS-2002	1.38 ± 1.34 1.0 (0.0–2.0)	1.38 + 1.39 1.0 (0.0–2.0)	*p* = 0.72^(U)^
Arterial hypertension; *n* (%)	1207 (81.06%)	480 (76.07%)	*p* = 0.009^(chi2)^
Diabetes; *n* (%)	410 (27.54%)	209 (33.12%)	*p* = 0.01^(chi2)^
Lipid disorders; *n* (%)	682 (45.77%)	231 (36.61%)	*p* = 0.001^(chi2)^
Stroke; *n* (%)	251 (16.85%)	113 (17.91%)	*p* = 0.63^(chi2)^
Coronary artery disease; *n* (%)	508 (34.09%)	238 (37.72%)	*p* = 0.11^(chi2)^
Myocardial infarction; *n* (%)	131 (8.79%)	101 (16.01%)	*p* < 0.001^(chi2)^
Artrial fibrillation; *n* (%)	314 (21.17%)	164 (26.11%)	*p* = 0.013^(chi2)^
Heart failure; *n* (%)	761 (51.01%)	299 (47.39%)	*p* = 0.12^(chi2)^
Obstructive lung diseases; *n* (%)	172 (11.54%)	81 (12.84%)	*p* = 0.39^(chi2)^
Osteoarthritis; *n* (%)	558 (37.45%)	159 (25.20%)	*p* < 0.001^(chi2)^
Osteoporosis; *n* (%)	436 (29.28%)	88 (13.95%)	*p* < 0.001^(chi2)^
Fractures; *n* (%)	285 (19.87%)	74 (12.19%)	*p* < 0.001^(chi2)^
Gastrointestinal diseases; *n* (%)	272 (18.26%)	138 (21.87%)	*p* = 0.053^(chi2)^
Neoplastic diseases; *n* (%)	212 (14.23%)	114 (18.07%)	*p* = 0.02^(chi2)^
Depression; *n* (%)	537 (36.04%)	158 (25.04%)	*p* < 0.001^(chi2)^
Dementia; *n* (%)	696 (46.90%)	246 (39.23%)	*p* < 0.001^(chi2)^
Pressure ulcers; *n* (%)	92 (6.37%)	51(8.35%)	*p* = 0.10^(chi2)^
Urine incontinence; *n* (%)	264 (18.40%)	77 (12.73%)	*p* = 0.001^(chi2)^

BMI: body mass index, (U): U Mann–Whitney Test, (chi2): chi-square test.

**Table 2 nutrients-15-01612-t002:** Comparison of NRS-2002 according to the prevalence of diseases.

Disease	Gender	NRS-2002				*p* Value
		In Patients with the Presence of Particular Disease		In Patients without Particular Disease		
		Mean ± SD	Median (Quartiles)	Mean ± SD	Median (Quartiles)	
Arterial hypertension	Women	1.34 ± 1.28	1.0 (0.0–2.0)	1.57 ± 1.53	1.0 (0.0–3.0)	*p* = 0.11
	Men	1.30 ± 1.31	1.0 (0.0–2.0)	1.64 ± 1.58	1.0 (0.0–3.0)	*p* = 0.05
Diabetes	Women	1.44 ± 1.29	1.0 (0.0–2.0)	1.36 ± 1.35	1.0 (0.0–7.0)	*p* = 0.15
	Men	1.39 ± 1.37	1.0 (0.0–2.0)	1.37 ± 1.40	1.0 (0.0–2.0)	*p* = 0.76
Lipid disorders	Women	1.07 ± 1.14	1.0 (0.0–2.0)	1.65 ± 1.43	1.0 (0.0–7.0)	*p* < 0.001
	Men	1.00 ± 1.20	1.0 (0.0–2.0)	1.60 ± 1.44	1.0 (0.0–3.0)	*p* < 0.001
Stroke	Women	1.70 ± 1.44	2.0 (0.0–3.0)	1.32 ± 1.30	1.0 (0.0–7.0)	*p* < 0.001
	Men	1.64 ± 1.48	1.0 (0.0–3.0)	1.32 ± 1.36	1.0 (0.0–2.0)	*p* = 0.03
Coronary artery disease	Women	1.32 ± 1.28	1.0 (0.0–6.0)	1.41 ± 1.36	1.0 (0.0–2.0)	*p* = 0.34
	Men	1.50 ± 1.40	1.0 (0.0–3.0)	1.31 ± 1.38	1.0 (0.0–2.0	*p* = 0.08
Myocardial infarction	Women	1.46 ± 1.35	1.0 (0.0–2.0)	1.37 ± 1.33	1.0 (0.0–2.0)	*p* = 0.45
	Men	1.44 ± 1.40	1.0 (0.0–2.0)	1.37 ± 1.38	1.0 (0.0–2.0)	*p* = 0.59
Atrial fibrillation	Women	1.66 ± 1.48	1.0 (0.0–3.0)	1.31 ± 1.29	1.0 (0.0–2.0)	*p* < 0.001
	Men	1.69 ± 1.45	2.0 (0.0–3.0)	1.26 ± 1.33	1.0 (0.0–2.0)	*p* < 0.001
Heart failure	Women	1.52 ± 1.36	1.0 (0.0–2.0)	1.23 ± 1.29	1.0 (0.0–2.0)	*p* < 0.001
	Men	1.57 ± 1.41	1.0 (0.0–3.0)	1.21 ± 1.34	1.0 (0.0–2.0)	*p* < 0.001
Obstructive lung diseases	Women	1.39 ± 1.35	1.0 (0.0–2.0)	1.38 ± 1.33	1.0 (0.0–2.0)	*p* = 0.93
	Men	1.49 ± 1.42	1.0 (0.0–3.0)	1.36 + 1.38	1.0 (0.0–2.0)	*p* = 0.45
Osteoarthritis	Women	1.15 + 1.25	1.0 (0.0–2.0)	1.49 ± 1.39	1.0 (0.0–2.0)	*p* < 0.001
	Men	1.15 ± 1.19	1.0 (0.0–2.0)	1.46 ± 1.44	1.0 (0.0–3.0)	*p* = 0.04
Osteoporosis	Women	1.20 ± 1.29	1.0 (0.0–2.0)	1.48 ± 1.36	1.0 (0.0–2.0)	*p* < 0.001
	Men	1.44 ± 1.46	1.0 (0.0–2.5)	1.37 ± 1.38	1.0 (0.0–2.0)	*p* = 0.82
Fractures	Women	1.48 ± 1.39	1.0 (0.0–2.0)	1.36 ± 1.32	1.0 (0.0–2.0)	*p* = 0.24
	Men	1.59 ± 1.44	1.0 (0.0–3.0)	1.36 ± 1.38	1.0 (0.0–2.0)	*p* = 0.18
Gastrointestinal diseases	Women	1.38 ± 1.27	1.0 (0.0–2.0)	1.38 ± 1.35	1.0 (0.0–2.0)	*p* = 0.73
	Men	1.53 ± 1.39	1.0 (0.0–3.0)	1.34 ± 1.38	1.0 (0.0–2.0)	*p* = 0.1
Neoplastic diseases	Women	1.48 ± 1.39	1.0 (0.0–2.0)	1.37 ± 1.33	1.0 (0.0–2.0)	*p* = 0.55
	Men	1.66 + 1.48	1.0 (0.0–3.0)	1.32 ± 1.36	1.0 (0.0–2.0)	*p* = 0.02
Depression	Women	1.33 ± 1.27	1.0 (0.0–2.0)	1.41 + 1.37	1.0 (0.0–2.0)	*p* = 0.91
	Men	1.35 ± 1.33	1.0 (0.0–2.0)	1.39 ± 1.41	1.0 (0.0–2.0)	*p* = 0.58
Dementia	Women	1.61 ± 1.39	1.0 (0.0–3.0)	1.19 ± 1.26	1.0 (0.0–2.0)	*p* < 0.001
	Men	1.68 ± 1.47	2.0 (0.0–3.0)	1.19 ± 1.29	1.0 (0.0–2.0)	*p* < 0.001
Pressure ulcers	Women	2.51 ± 1.58	3.0 (1.0–3.0)	1.30 ± 1.28	1.0 (1.0–2.0)	*p* < 0.001
	Men	2.54 ± 1.61	3.0 (0.0–4.0)	1.28 ± 1.31	1.0 (0.0–2.0)	*p* < 0.001
Urine incontinence	Women	0.95 ± 1.17	0.5 (0.0–2.0)	1.46 ± 1.35	1.0 (0.0–2.0)	*p* < 0.001
	Men	0.80 ± 1.27	0.0 (0.0–2.0)	1.46 ± 1.38	1.0 (0.0–2.0)	*p* < 0.001

SD: standard deviation; compared by U Mann–Whitney test.

**Table 3 nutrients-15-01612-t003:** Comparison of age according to the prevalence of diseases.

Disease	Gender	Age				*p* Value
		In Patients with the Presence of Particular Disease		In Patients without Particular Disease		
		Mean ± SD	Median (Quartiles)	Mean ± SD	Median (Quartiles)	
Arterial hypertension	Women	82.77 ± 7.43	84 (79–88)	80.90 ± 8.40	82 (74–88)	*p* = 0.001
	Men	81.89 ± 8.28	83 (78–87)	80.78 ± 8.85	82 (74–87)	*p* = 0.24
Diabetes	Women	82.02 ± 7.54	83 (83–87)	82.57 ± 7.70	84 (79–88)	*p* = 0.13
	Men	80.96 ± 8.11	82 (75–88)	81.95 ± 8.56	84 (77–88)	*p* = 0.17
Lipid disorders	Women	80.61 ± 7.62	82 (76–86)	83.94 ± 7.36	85(81–89)	*p* < 0.001
	Men	79.59 ± 8.67	81 (73–86)	82.80 ± 8.06	84 (79–88)	*p* < 0.001
Stroke	Women	83.07 ± 7.59	84 (79–88)	82.28 ± 7.67	83 (78–88)	*p* = 0.23
	Men	80.81 ± 8.81	82 (74–88)	81.80 ± 8.33	83 (77–88)	*p* = 0.26
Coronary artery disease	Women	83.96 ± 6.99	85 (80–88)	81.61 ± 7.98	83 (77–87)	*p* < 0.001
	Men	83.70 ± 7.52	85 (80–89)	80.37 ± 8.70	81 (74–87)	*p* < 0.001
Myocardial infarction	Women	83.70 ± 7.79	85 (80–88)	82.29 ± 7.64	83 (78–88)	*p* = 0.02
	Men	83.25 ± 7.59	85 (79–88)	81.31 ± 8.54	83 (75–88)	*p* = 0.06
Atrial fibrillation	Women	84.14 ± 6.59	85 (81–89)	81.99 ± 7.83	83 (77–88)	*p* < 0.001
	Men	84.90 ± 7.19	86 (82–89	80.54 ± 8.43	82 (74–87)	*p* < 0.001
Heart failure	Women	84.17 ± 6.76	85 (80–89)	80.58 ± 8.10	82 (75–87)	*p* < 0.001
	Men	83.68 ± 7.87	85 (80–89)	79.77 ± 8.48	81 (74–86)	*p* < 0.001
Obstructive lung diseases	Women	82.37 ± 6.95	83 (78–87)	82.17 ± 7.99	83(78–88)	*p* = 0.54
	Men	82.02 ± 7.72	84 (78–87)	82.42 + 7.52	84 (78–88)	*p* = 0.72
Osteoarthritis	Women	82.30 ± 7.37	83 (78–87)	82.48 ± 7.83	84 (78–88)	*p* = 0.34
	Men	81.61 ± 8.54	82 (76–88)	81.63 ± 8.37	83 (75–88)	*p* = 0.89
Osteoporosis	Women	82.76 ± 6.95	84 (80–87)	82.27 ± 7.93	83 (78–88)	*p* = 0.52
	Men	83.82 ± 7.31	85 (79–89)	81.27 ± 8.54	82(75–88)	*p* = 0.008
Fractures	Women	84.31 ± 6.30	85 (81–88)	81.91 ± 7.87	83 (77–88)	*p* < 0.001
	Men	82.77 ± 9.71	84 (76–91)	81.45 ± 8.14	83 (76–87)	*p* = 0.11
Gastrointestinal diseases	Women	81.41 ± 7.86	83 (77–87)	82.64 ± 7.60	84 (79–88)	*p* = 0.01
	Men	81.62 ± 7.88	83 (78–87)	81.63 ± 8.58	83 (75–88)	*p* = 0.96
Neoplastic diseases	Women	81.69 ± 8.11	83 (76–88)	82.54 ± 7.58	84 (79–88)	*p* = 0.17
	Men	83.11 ± 8.00	84 (79–89)	81.30 ± 8.49	82 (75–88)	*p* = 0.03
Depression	Women	82.21 ± 7.02	83 (78–87)	82.53 ± 7.99	84 (78–88)	*p* = 0.27
	Men	81.15 ± 8.11	82 (75–87)	81.78 ± 8.53	83 (76–88)	*p* = 0.06
Dementia	Women	84.59 ± 6.61	86 (81–89)	80.49 ± 8.02	82 (75–86)	*p* < 0.001
	Men	83.80 ± 7.52	84 (79–89)	80.22 ± 8.68	82 (74–87)	*p* < 0.001
Pressure ulcers	Women	84.26 ± 8.17	85 (80–90)	82.23 ± 7.63	83 (78–88)	*p* = 0.01
	Men	83.82 ± 6.36	85 (82–88)	81.44 ± 8.50	82 (75–88)	*p* = 0.06
Urine incontinence	Women	81.86 ± 7.54	83 (78–87)	82.46 ± 7.69	84 (78–88)	*p* = 0.14
	Men	82.15 ± 8.55	84 (76–89)	80.92 ± 9.90	83 (74–89)	*p* = 0.45

SD: standard deviation; compared by U Mann–Whitney test.

## Data Availability

The statistical data used to support the presented findings may be obtained upon request to corresponding author.

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
