# Peer review of "The Relationship between Nutritional Risk and the Most Common Chronic Diseases in Hospitalized Geriatric Population from Central Poland"

_nutrients, 2023, doi:10.3390/nu15071612_

Round 1

Reviewer 1 Report

abstract: remove "subjects" and replace with "participants - this is not an intervention study

results:

ln 106, add SD after age and present years with decimal

table 1. commas used in place of period in some areas of the table resulted in substantial, initial confusion 

table 3 can be removed since authors established differences in disease by sex (table 1)

I am curious why authors did not present differences in NRS-2002 between sexes. This seems relevant to your research aim in addition to disease by sex.

Results presentation of OR data in figures needs improvement. Figures need footnotes for abbreviations. Periods in place of commas. I don't generally follow the data using 0 rather than 1.0 for nonsignificance with OR and CI but will defer that to the editorial team. Resource for authors: Odds Ratio - StatPearls - NCBI Bookshelf (nih.gov)

ln 143 Arterial needs to be added before hypertension in 

Discussion

Overall impression is a lot of non-relevant information is provided with little useful interpretation of the current study findings relative to the literature presented. See ln 156-185 as example. Even in lns 204-210 when study results are mentioned, the discussion does not bring forward useful data/information from the present study. The Discussion section needs a full revision in my opinion. It is not effective as presented. Another example of issues with the Discussion section is ln 262-263 - this statement is not supported by the evidence of the present study results in my opinion.

Conclusion

Recommend revision of first sentence. Such as, this study revealed which medical conditions coexist with increased nutrition risk in a hospitalized geriatric population. These relationships vary by sex.

For the second sentence, revise from "older subject" to "geriatric patient."

Author Response

Comments and Suggestions for Authors

abstract: remove "subjects" and replace with "participants - this is not an intervention study

Response: "subjects" has been replaced with "participants”

results:

ln 106, add SD after age and present years with decimal

Response: has been done

table 1. commas used in place of period in some areas of the table resulted in substantial, initial confusion 

Response: has been corrected

table 3 can be removed since authors established differences in disease by sex (table 1)

Response: Table 3 shows the differences in age in relation to given presented diseases. As several diseases were reported more often in older age (what is not unanticipated), some were related to younger age. This phenomenon has been further talked over in the discussion section of the study. To present data more clearly, the differences were presented in relation to sex.

I am curious why authors did not present differences in NRS-2002 between sexes. This seems relevant to your research aim in addition to disease by sex.

Response: has been presented in the Table 1. NRS-2002 was virtually the same in women and men.

Results presentation of OR data in figures needs improvement. Figures need footnotes for abbreviations. Periods in place of commas. I don't generally follow the data using 0 rather than 1.0 for nonsignificance with OR and CI but will defer that to the editorial team. Resource for authors: Odds Ratio - StatPearls - NCBI Bookshelf (nih.gov)

Response: The presentation of figures has been modified to present more clearly the association of particular diseases to the nutritional risk. Abbreviations have been explained. Periods in place of commas have been corrected.

ln 143 Arterial needs to be added before hypertension in 

Response: has been added as suggested.

Discussion

Overall impression is a lot of non-relevant information is provided with little useful interpretation of the current study findings relative to the literature presented. See ln 156-185 as example. Even in lns 204-210 when study results are mentioned, the discussion does not bring forward useful data/information from the present study. The Discussion section needs a full revision in my opinion. It is not effective as presented. Another example of issues with the Discussion section is ln 262-263 - this statement is not supported by the evidence of the present study results in my opinion.

Response: Lines 156-185: We intended to show the role of NRS-2002 in nutritional risk screening and to justify the more in depth the topic of our study. Nevertheless, we do agree that this literature review may be to extensive and we have importantly modified and shortened this part of the discussion.

            Lines 204-210: have also been modified.

            Lines 262-263: the relationship of NRS-2002 to different accompanying diseases, and association of those diseases to the age of the participants, have been presented more clearly. The relative significance of these accompanying conditions to the nutritional risk has been discussed.

Conclusion

Recommend revision of first sentence. Such as, this study revealed which medical conditions coexist with increased nutrition risk in a hospitalized geriatric population. These relationships vary by sex.

Response: Conclusions have been modified for better clarity of presentation. Some relationship to sex has been tackled in the discussion. Nevertheless, as shown in the Table 2 and Figures 1-3, we believe that the relationship of nutritional risk to coexisting conditions is generally very similar in women and men.

For the second sentence, revise from "older subject" to "geriatric patient."

Response: Here we would like to ask editorial office for a decision concerning the use of some terms. Our experience is that the term “older subjects” is usually considered neutral while “geriatric patients” as “elderly” have been criticized for their pejorative meaning.

Reviewer 2 Report

An interesting work titled "The relationship between nutritional status and the most common chronic diseases in hospitalized geriatric population from  Central Poland" however, the authors should explain:

- justify the research in detail, as the Authors write that there are many works in this field, without citing them in the Introduction

- in methodology the Authors did not clarify whether all patients in the study were ill. In Table 2, patients are divided into diseased and disease-free. Whether without disease, that is, without said disease or with another disease (and what kind); The answer to this question may decide whether to accept or reject the study.

- in logistic regression what was the OR=1 reference

- in the results, the Authors overgeneralize - the results apply only to the study group. In addition, the study did not assess the effect of excessive BMI on diseases - what can be considered a defect in the study; perhaps a reference to BMI, below and above the recommended values, would be more appropriate to assess the association with disease as well as age – these issues remain unresolved in the study. Some diseases can be related to malnutrition, but also in large part to overfeeding - which has not been studied.

In this context, the title of the work should also be verified.

- the discussion is too long and chaotic, there is no leading thought, unclear strengths and weaknesses of the study

- the Authors in the work sometimes confuse the concepts of nutritional assessment with nutritional status.

Author Response

Comments and Suggestions for Authors

An interesting work titled "The relationship between nutritional status and the most common chronic diseases in hospitalized geriatric population from  Central Poland" however, the authors should explain:

- justify the research in detail, as the Authors write that there are many works in this field, without citing them in the Introduction.

Response: Additional citations have been added to the Introduction section.  Furthermore, we reformulated the justification of our study. The mail goal is to achieve the investigation of  the nutritional risk in light of most common diseases occurring in older age. The literature provides relatively wide range of publications referring to a single disease or disorders, and lacking in the studies considering the whole patients’ anamnesis.

- in methodology the Authors did not clarify whether all patients in the study were ill. In Table 2, patients are divided into diseased and disease-free. Whether without disease, that is, without said disease or with another disease (and what kind); The answer to this question may decide whether to accept or reject the study.

Response: The survey was conducted in the “real-life”  population of older people hospitalized in geriatric ward. As the patients were referred for hospitalization, all were ill. The lack of one particular disease would not mean the presence of another. Typically for their age, our subjects presented often with multiple comorbidities.

The prevalence of the concomitant diseases has been presented in the Table 1. In the Table 2 and 3, we amended the headlines – to indicate that patients were divided according to the rule of presence or absence of one particular disease. The data have been provided separately for men and women.

- in logistic regression what was the OR=1 reference

Response:  OR=1 was NRS-2002 < 3 points (no nutritional risk). The presentation of figures has been modified to present more clearly the association of particular diseases to the nutritional risk.

- in the results, the Authors overgeneralize - the results apply only to the study group. In addition, the study did not assess the effect of excessive BMI on diseases - what can be considered a defect in the study; perhaps a reference to BMI, below and above the recommended values, would be more appropriate to assess the association with disease as well as age – these issues remain unresolved in the study. Some diseases can be related to malnutrition, but also in large part to overfeeding - which has not been studied.

The fact that the results apply to study group and may be different in other settings has been acknowledged in the shortcomings of the study.

Concerning overfeeding (obesity) we would like to draw attention to the two important points:

1) In hospitalized geriatric patients the prevalence of obesity is not high (less than 4% of the patients had BMI > 35).

2) NRS-2002 is connected with BMI (contains BMI in its definition), and further adjustment for BMI may lead calculations to be biased.

Nevertheless, after taking into account the Reviewer’s suggestions, we performed dichotomization of BMI, on 4 different levels of BMI. For BMI cut-off points of 23, 25, 27 and 30 kg/m2, the positive relationship between these two variables is extremely strong (chi square <0.001 for all BMI cut-offs against dichotomized NRS-2002). 

When we put BMI into regression models, it turned out that variable was such a strong predictor of NRS that only some diseases were still present in the model. Pressure ulcers, atrial fibrillation and dementia (borderline significance) were related to higher nutritional risk, while lipid disorders, urine incontinence and osteoarthritis (only in one model) were related to lower nutritional risk

These additional considerations have been provided in the results section of the study.

In this context, the title of the work should also be verified.

Response: Thank you for that comment. The title has been modified: “nutritional status” has been changed to “nutritional risk”.

- the discussion is too long and chaotic, there is no leading thought, unclear strengths, and weaknesses of the study.

Response: Presented research focuses on the association between NRS-2002 and a variety of concomitant diseases among hospitalized older people. The results indicate that there is a bidirectional association between nutritional risk and disorders typical for geriatric patients. Some, like lipid disorders, osteoarthritis, or urine incontinence are related to better NRS-2002. On the other hand, presence of stroke, atrial fibrillation, dementia, and pressure ulcers are associated with increased risk of malnutrition. The main strength of our study is the number of subjects and assessment of wide variety of geriatric disorders.

According to the suggestions of the Reviewer the discussion has been reshaped to better address these study points.

The most important limitations of the study have been mentioned in the last paragraph of the discussion.

- the Authors in the work sometimes confuse the concepts of nutritional assessment with nutritional status.

Response:   Thank you for drawing our attention to that matter. We made changes in the manuscript to thoroughly distinguish nutritional status and nutritional assessment.

Reviewer 3 Report

The authors present an interesting article. They aimed to analyze the relationship between Nutrition Risk Screening 2002 (NRS-2002) and the prevalence of concomitant chronic diseases among hospitalized older adults. Authors found that nutritional risk was associated with stroke, atrial fibrillation, heart failure, dementia and pressure ulcers. The manuscript consists of 13 pages, including 3 tables, 3 figures and 66 references.

The topic of the study is highly relevant in geriatric departments and everyday clinical practice.

Comments and suggestions to the manuscript:

Abstract:

-          In the abstract authors gave an explanation for the association between comorbidities and NRS score: “The explanation of this phenomenon may be the occurrence of some diseases in relatively earlier age, mild manifestation of some disorders and adaptation to chronic ailments with less frequent reporting at advanced age.” This conclusion is not clear in the context of the aforementioned results and needs further explanation.

Keywords and introduction:

-          The authors should avoid the term “elderly”. It is better to use “older adults” or “older people”.

-          Overall, the introduction is detailed and informative to the reader.

Materials and Methods

Section statistical analysis

-          Normal distribution was analysed via Shapiro wilk test. Please give an explanation why all data irrespective normal distribution or non-normal distribution was evaluated using the Mann Whitney U test comparing unpaired samples. For interval-scaled and normally distributed data t-test for unpaired samples might be more adequate and has more power.

Results

-          Presentation of results in tables is insufficient and some formal points need correction.

-          The use of a comma or a dot in table 1 is inconsistent. The authors should use consistently a dot as decimal separator.

-          The authors should state all p-values in the tables instead of “ns”.  

-          Please explain why authors show some data in bold in table 3

-          Abbreviations in figure 1-3 should be explained (for example OR, UQ and LQ)

Discussion

-          Discussion is detailed and informative.

-          I appreciate authors’ clear statement on study limitations.

Author Response

Comments and Suggestions for Authors

The authors present an interesting article. They aimed to analyze the relationship between Nutrition Risk Screening 2002 (NRS-2002) and the prevalence of concomitant chronic diseases among hospitalized older adults. Authors found that nutritional risk was associated with stroke, atrial fibrillation, heart failure, dementia and pressure ulcers. The manuscript consists of 13 pages, including 3 tables, 3 figures and 66 references.

The topic of the study is highly relevant in geriatric departments and everyday clinical practice.

Comments and suggestions to the manuscript:

Abstract:

-          In the abstract authors gave an explanation for the association between comorbidities and NRS score: “The explanation of this phenomenon may be the occurrence of some diseases in relatively earlier age, mild manifestation of some disorders and adaptation to chronic ailments with less frequent reporting at advanced age.” This conclusion is not clear in the context of the aforementioned results and needs further explanation.

Response:  In the Table 3 we placed the information about the mean age of patients with and without particular disease. We wanted to consider two things:

  1. Some diseases may be more associated with younger/older age. Younger subjects, statistically with better nutritional status as compared to older peers, more often report some diseases, e.g. lipid disorders or gastrointestinal diseases. As they present also statistically with better nutritional status as compared to older peers, this impact their interrelationship and may lead to spurious conclusions of protective role of those diseases as to nutritional status. In opposition to that, several diseases occur typically in older subjects, such as heart failure or pressure ulcers.
  2. Second perspective refers to age itself. We couldn’t employ age into logistic regression, as this variable is strictly connected with the NRS results itself. That would lead our analysis into bias.

Therefore, with awareness of those two factors, we wanted to highlight the role of age as the factor modifying our results. According to the suggestions of the Reviewer the presentation of these data, discussion and conclusions have been modified. 

Keywords and introduction:

-          The authors should avoid the term “elderly”. It is better to use “older adults” or “older people”.

 Response: Thank you for that comment. Relevant changes have been performed.

-          Overall, the introduction is detailed and informative to the reader.

 Thank you for acknowledging the quality of the introduction section.

Materials and Methods

Section statistical analysis

-          Normal distribution was analysed via Shapiro wilk test. Please give an explanation why all data irrespective normal distribution or non-normal distribution was evaluated using the Mann Whitney U test comparing unpaired samples. For interval-scaled and normally distributed data t-test for unpaired samples might be more adequate and has more power.

Response: We do agree that for normally distributed data t-test for unpaired samples has more power. Nevertheless, the majority of variables were either not normally distributed or did not met the criteria of homogeneity of variance. Hence, the quantitative variables were compared using a Mann-Whitney U-test. As some median and quartiles values overlapped even with statistical difference, for the clarity of presentation data have been expressed both as mean ± standard deviation (SD) and median (25%-75% quartiles).

Results

-          Presentation of results in tables is insufficient and some formal points need correction.

Response: Presentation of the tables has been modified and corrected according to the suggestions below.

-          The use of a comma or a dot in table 1 is inconsistent. The authors should use consistently a dot as decimal separator.

 Response: Table 1 was amended. Dots are used as decimal separators.

-          The authors should state all p-values in the tables instead of “ns”.  

Response: All p values have been added.

-          Please explain why authors show some data in bold in table 3

Response: Presentation has been corrected and unified.

-          Abbreviations in figure 1-3 should be explained (for example OR, UQ and LQ)

Response: The OR refers to Odds Ratio, LQ-Lower Quartile; UQ-Upper Quartile. Relevant amendments were done. The presentation of figures has been modified to present more clearly the association of particular diseases to the nutritional risk.

Discussion

-          Discussion is detailed and informative. 

-          I appreciate authors’ clear statement on study limitations.

 Thank you for the positive assessment of our work.

Round 2

Reviewer 1 Report

The revised manuscript is much improved from the original submission. This is an important study and now well-represented in the original article.

Reviewer 2 Report

Accept in present form.

Reviewer 3 Report

The authors of the manuscript "The relationship between nutritional risk and the most common chronic diseases in hospitalized geriatric population from Central Poland" addressed my points of criticism. The manuscript has been substantially improved. I have no further questions.